# Laboratory Evaluation of a SARS-CoV-2 RT-LAMP Test

**DOI:** 10.3390/tropicalmed8060320

**Published:** 2023-06-13

**Authors:** Sandra Menting, Annette Erhart, Henk D. F. H. Schallig

**Affiliations:** 1Amsterdam University Medical Centres, Academic Medical Centre at the University of Amsterdam, Laboratory for Experimental Parasitology, Department of Clinical Microbiology and Infection Prevention, Meibergdreef 9, 1105 AZ Amsterdam, The Netherlands; s.menting@amsterdamumc.nl; 2MRC Unit The Gambia at the LSHTM, Atlantic Boulevard, Fajara, Banjul P.O. Box 273, The Gambia; annette.erhart@lshtm.ac.uk

**Keywords:** COVID-19, RT-LAMP, RT-PCR, molecular diagnostic, performance, LMICs

## Abstract

There is a need to have more accessible molecular diagnostic tests for the diagnosis of severe acute respiratory syndrome coronavirus 2 disease in low- and middle-income countries. Reverse transcription loop-mediated isothermal amplification (RT-LAMP) may provide an attractive option as this technology does not require a complex infrastructure. In this study, the diagnostic performance of a SARS-CoV-2 RT-LAMP was evaluated using RT-PCR-confirmed clinical specimens of COVID-19-positive (n = 55) and -negative patients (n = 55) from the Netherlands. The observed sensitivity of the RT-LAMP test was 97.2% (95% CI: 82.4–98.0%) and the specificity was 100% (95% CI: 93.5–100%). The positive predictive value of the RT-LAMP was 100%, the negative predictive value 93.2% (95% CI: 84.3–97.3%), and the diagnostic accuracy was 96.4% (95% CI: 91.0–99.0%). The agreement between the RT-LAMP and the RT-PCR was “almost perfect” (κ-value: 0.92). The evaluated RT-LAMP might provide an attractive alternative molecular diagnostic tool for SARS-CoV-2 in resource limited settings.

## 1. Introduction

Many low- and middle-income countries (LMICs) have a fragile health system and the latter are under even greater pressure since the COVID-19 pandemic [1]. There is a need to establish and improve diagnostic and disease surveillance capacity in LMICs. With limited material and human resources available, accurate, simple, and rapid diagnostic methods for virus detection are needed to test as many individuals as possible, to promptly isolate confirmed cases, and to support rapid contact tracing for the surveillance and understanding of the local epidemiology [1]. The reference standard for the detection of severe acute respiratory syndrome coronavirus 2 (SARS-CoV-2) infection is a real-time reverse transcription polymerase chain reaction (real-time RT-PCR) [2]. However, the implementation of real-time RT-PCR requires specialized laboratories, infrastructure and personnel, and costly reagents and equipment, which are often not widely available in LMICs. Antigen detection based rapid diagnostic tests (Ag-RDTs) represent an attractive alternative, because of their low cost and ease of operation, but concerns about their sensitivity limit their use at particularly low viral loads [3,4,5,6,7,8,9].

The development of loop-mediated isothermal amplification (LAMP) represents an interesting advance in nucleic acid-based diagnostics [10,11] with several advantages. First, the amplification reaction is isothermal (between 60 and 65 °C) and does not require the use of a thermocycler. Second, the specificity of the reaction is high because of the design of four to six primers recognizing distinct sequences on the target. Third, the product can be visualized directly using simple detection methods, from fluorescence to colorimetric to lateral flow assay. Reverse transcription loop-mediated isothermal amplification (RT-LAMP) assays have now also been developed for the diagnosis of SARS-CoV-2 [10,11]. RT-LAMP combines the advantages of Ag-RDTs as they are easy, rapid, and do not require specialized infrastructures [10] or real-time RT-PCR (high sensitivity) [10,11,12,13]. These properties make RT-LAMP an attractive candidate diagnostic test for implementation in LMICs. However, the diagnostic performance of such a test needs to be independently established under controlled laboratory conditions before implementation.

The aim of the current study was to evaluate the performance of the SARS-CoV-2 RT-LAMP kit (Coris BioConcept, Gembloux, Belgium; CE marked) against gold standard real-time RT-PCR using both positive and negative clinical samples of the SARS-CoV-2 virus.

## 2. Materials and Methods

### 2.1. Ethical Statement

The use of clinical specimens from the Amsterdam University Medical Centre (Amsterdam UMC) was approved by the Biobank Review Committee (approval: 2021.0259) for the protocol: “Enhancing Diagnostic & Surveillance Capacity in Low-Middle Income Countries.” Informed consent was obtained before the storage and usage of residual materials from COVID-19 patients. Ethical review was waved for the anonymized use of stored diagnostic specimens for diagnostic evaluation in accordance with Dutch law.

### 2.2. Clinical Samples

The clinical samples comprised nasopharyngeal swabs from 110 Dutch suspected cases of COVID-19 (based on clinical symptoms) stored at the departmental biobank of the department of Medical Microbiology and Infection Prevention of the Amsterdam UMC. Specimens were obtained as part of the routine clinical diagnostic practice for SARS-CoV-2 in place at Amsterdam UMC [14]. Nasopharyngeal swabs were taken from patients admitted at AMSTERDAM UMC with the clinical suspicion of having COVID-19 diseases. All had respiratory complaints and were hospitalized (not to the IC unit) during the peak of the pandemic in the Netherlands. Nasopharyngeal swabs were collected in a 3 mL UTM viral transport medium (COPAN ITALIA spa Brescia, Italy). In total, 110 swabs were available for analysis. Cases were confirmed as being positive or negative using an established SARS-CoV-2 real-time RT-PCR targeting the E-gene with a confirmed sensitivity of >95 and specificity of 100%, which was performed by complying to the established protocol [2].

### 2.3. Sample Size Calculation

The RT-LAMP test under evaluation should have a minimally acceptable sensitivity and specificity of 97%. To make a precise estimation of the sensitivity and specificity, the desired 95% confidence interval is ±5%. With these assumptions, the number of samples to be analysed can be estimated following WHO/TDR guidelines for the evaluation of diagnostics [15]: p ± *z* × √[p(1 − p)/n]
where p = sensitivity (or specificity) measured as a proportion, n = number of samples from infected people (or for specificity from non-infected) and *z* = 1.96 (if we use a 95% confidence interval).

According to this equation, at least 44 SARS-CoV-2 positive or negative samples should be included to be able to determine with 95% confidence, as recommended by the Standards for Reporting Diagnostic accuracy studies [16], if the sensitivity or specificity is 97% ± 5%.

### 2.4. Procedures

SARS-CoV-2 RNA was extracted using an EMag (bioMérieux SA, Marcy-l′Étoile, France). Next, a SARS-CoV-2 real-time RT-PCR (gold standard for this study) targeted at the E-gene, which was performed according to a previously established protocol [2], was performed on all 110 clinical samples. Samples with a Ct value > 37.0 were considered to be negative.

Subsequently, a Reverse transcription loop-mediated isothermal amplification (SARS-CoV-2 RT-LAMP) assay (Coris BioConcept, Gembloux, Belgium; Lot: 47980H2208; expiration date: 20 October 2022) utilizing a primer mix targeting the SARS-CoV-2 ORF1ab region, the N gene, and the E gene was used as the test under investigation. The samples were subjected to RT-LAMP without having the final results of the gold standard real-time RT-PCR available. RT-LAMP was performed according to the manufacturer’s instructions. The RT-LAMP assay is based on fluorometric detection using a DNA intercalating agent. The RT-LAMP assay was performed at 63 °C for 30 min using a CFX-96 (Bio-Rad, Veenendaal, The Netherlands) real-time thermocycler. The RT-LAMP conditions consisted of a temperature of 63 °C, maintained during 30 cycles of 60 s each for a total running time of 30 min. The fluorescence signal was collected at each of the 30 repeats. The fluorescence channel to use for the dye was the SYBR Green/FAM channel (450–490 nm for excitation, 510–530 nm for detection). The volume of the reaction mix including the sample was 20 μL (5 μL of extracted RNA or 5 μL of positive control to 15 μL of reaction mix). The test was performed in 0.2 mL (PCR) tubes. 

A positive control provided in the kit was used to confirm that the test was performed with effective reagents and in correct experimental conditions. A negative control (No Template Control, NTC) was performed with 5 μL of molecular biology grade water as the sample. The positive control was analysed as positive if it displayed a fluorescence growth curve with a Ct value < 30. The negative control could not display a fluorescence growth curve.

### 2.5. Data Analysis

The performance of the RT-LAMP was evaluated against the gold standard real-time RT-PCR, using both results to calculate the sensitivity, specificity, and negative and positive predictive value of the RT-LAMP using the MedCalc Software Ltd. diagnostic test evaluation calculator, https://www.medcalc.org/calc/diagnostic_test.php (Version 20.013; accessed on 30 December 2022). The agreement between the RT-LAMP and real-time RT-PCR was determined by calculating the kappa (k) value with 95% confidence intervals using GraphPad software, https://www.graphpad.com/quickcalcs/ (Version 12/2022; accessed on 30 December 2022).

## 3. Results

The 110 samples were first analysed by gold standard SARS-CoV-2 real-time RT-PCR and subsequently by RT-LAMP. PCR revealed that there were 55 cases positive and 55 cases negative for SARS-CoV-2. The positive SARS-CoV-2 PCR samples had a mean Ct value of 24.0 (range: 15.0–33.7). Nine of these samples had a Ct value > 30.0.

All clinical samples were next tested by RT-LAMP (operator was blinded from the PCR results) and these results are presented in Table 1. Positive and negative controls were included in each test and there were no amplification deviations observed.

In total, 51 out of 55 real-time RT-PCR positive samples were also found positive with the RT-LAMP test, while 4 samples tested negative. The latter were found to have relatively low viral loads as their cycle threshold (Ct) values ranged from 30.54 to 33.66. However, 5 other samples with a Ct value > 30.0 were found positive with RT-LAMP. All 55 negative real-time RT-PCR samples were also found negative by RT-LAMP. 

The observed sensitivity of the RT-LAMP test was 97.2% (95% CI: 82.4–98.0%) and the specificity was 100% (95% CI: 93.5–100%). The positive predictive value of the RT-LAMP was 100% and the negative predictive value was 93.2% (95% CI: 84.3–97.3%). The diagnostic accuracy (i.e., the overall probability that a patient is correctly classified) of the RT-LAMP was 96.4% (95% CI: 91.0–99.0%).

The agreement between the RT-LAMP and the real-time RT-PCR was considered “almost perfect,” with a κ-value of 0.93 (95% CI: 0.86–0.99; SE of kappa = 0.04).

## 4. Discussion

The observed diagnostic performance of the RT-LAMP test under evaluation (sensitivity 97.2%; specificity 100%) exceeded the WHO recommendations for Ag-RDTs, which should have a minimal sensitivity of 80% and a minimal specificity of 97% [6]. The RT-LAMP test also met the sample size assumptions of having a sensitivity and specificity of >97%. In total, 4 positive RT-PCR samples were not detected by RT-LAMP, probably due to their low viral load, hence confirming prior reports showing a slightly lesser sensitivity compared to RT-LAMP for the diagnosis of SARS-CoV-2[10,11]. 

On the other hand, the RT-LAMP was able to detect SARS-CoV-2 in 5 positive samples with a Ct > 30. These might have been missed by Ag-RDTs since the latter often have a poorer performance when testing samples with Ct values > 25 [17] and thus, these might have been missed while using an antigen detection test and this underpins the added value of the RT-LAMP.

A limitation of the current evaluation is the fact that we did not specifically assess the diagnostic performance of the RT-LAMP on the clinical specimens of patients with other confirmed lung diseases. However, all tested samples were collected at the height of the COVID-19 pandemic in the Netherlands from cases with clinical suspicion of disease caused by SARS-CoV-2, as all had respiratory symptoms and can, as such, be considered a representative sample.

The present study confirms the good diagnostic performance of the RT-LAMP under evaluation as compared to real-time RT-PCR, with the advantages of having much less technical and costs requirements than real-time RT-PCR. Moreover, there are several options to further simplify the RT-LAMP procedure to enable its implementation as a molecular diagnostic in resource-limited settings. First, there is no need to use a sophisticated real-time machine for amplification. As the amplification reaction is isothermal, it can be performed in a simple Loopamp incubator (Eiken Chemical Co., Tokyo, Japan) [18]. Even the use of water baths has been proposed to facilitate the isothermal amplification process. Secondly, the current protocol uses fluorescence detection via a real-time PCR machine. This can be circumvented by visualising the amplification results under the illumination of UV light, which can be observed by the naked eye [18]. A slightly more sophisticated read-out method is the use of lateral flow-based strips for the detection of amplified SARS-CoV-2 viral mRNA [19,20]. A further simplification could be sought by using less sophisticated (or even non) nucleic acid extraction methods [21], but these will need further validation, particularly for RNA.

## 5. Conclusions

The RT-LAMP for SARS-CoV-2 evaluated in the present study has good diagnostic performance (sensitivity: 97.2%; specificity: 100%) compared to the gold standard reference, real-time RT-PCR. With some adaptations, such as simplifying the results read-out, this assay could be implemented as a simple molecular diagnostic tool in resource-limited settings.

## Figures and Tables

**Table 1 tropicalmed-08-00320-t001:** Comparison of diagnostic testing with real-time RT-PCR (gold standard) and RT-LAMP on RNA samples isolated from nasopharyngeal swabs collected from COVID-19-positive and -negative patients.

RT-LAMP	Real-time RT–PCR			
		Positive	Negative	Total
	Positive	51	0	51
	Negative	4	55	59
		55	55	110

## Data Availability

The data that support the findings of this study are contained within this article.

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
