# Peer review of "Laboratory Evaluation of a SARS-CoV-2 RT-LAMP Test"

_tropicalmed, 2023, doi:10.3390/tropicalmed8060320_

Round 1

Reviewer 1 Report

I have been asked to review a paper entitled ‘Laboratory evaluation of a SARS-CoV-2 RT-LAMP test 2’. Herein are some selective comments towards the authors that might improve their work.

Introduction

English improvement is needed. Several typos were found (e.g. lines 45-48).

Please define RT-LAMP better, as it came after the description of the LAMP.

There are already commercially available RT-LAMP for COVID detection; please address this information and what is the novelty of the work compared to the commercially available devices.

Results and Discussion

Please consider adding the Ct values for all samples.

Again, English improvement is needed.

Please look at this recently published reference that has a lot of helpful information https://doi.org/10.3390/diagnostics12092232

Conclusion

Please add a final statement about how you see the device being used as an alternative to RT-PCR.

Also, please add the sensitivity and specificity %. 

Author Response

We thank the reviewer for his/her time to consider the manuscript and to provide us with the valuable comments. We have addressed the points raised by the reviewer below as follows:

Introduction

English improvement is needed. Several typos were found (e.g. lines 45-48).

Response: the errors in these lines have been changed.

Please define RT-LAMP better, as it came after the description of the LAMP.

Response:  we have looked at this and we think we have appropriately presented the RT-LAMP in the introduction.

There are already commercially available RT-LAMP for COVID detection; please address this information and what is the novelty of the work compared to the commercially available devices.

Response: indeed several commercial RT-LAMP tests are available, but as with all tests an independent laboratory evaluation is needed to ensure the quality of such a test before implementation. This is the reason why we conducted this work. We have now mentioned this  in lines 48-49: “However, the diagnostic performance of such a test needs to be independently established under controlled laboratory conditions before implementation.”

Results and Discussion

Please consider adding the Ct values for all samples.

Response:  the mean Ct value (with range) has now been provided in the Results section.

Again, English improvement is needed.

Response: these have now been corrected

Please look at this recently published reference that has a lot of helpful information https://doi.org/10.3390/diagnostics12092232

Response: this is indeed a valuable manuscript and we carefully looked at it to see how we can further improve some of the information in our publication. In particular on reporting the Ct values.

Conclusion

Please add a final statement about how you see the device being used as an alternative to RT-PCR.

Response: statement has been added.

Also, please add the sensitivity and specificity %. 

Response: sensitivity and specificity of LAMP have been added in the conclusion

Reviewer 2 Report

In this manuscript Menting et al. evaluated an RT-LAMP assay for SARS-CoV-2. There are numerous papers describing the performance of RT-LAMP compared to RT-PCR, considering its advantages and limitations. The major shortcoming of this report is that it does not add relevant new information to the field. The authors state that there is a need for more accessible molecular tests for diagnostics in low- and middle-income countries with limited resources. Why then rely on extracted RNA rather than direct sample testing, which is possible with LAMP? What is the advantage of this protocol over PCR? The authors calculated that the sensitivity of RT-LAMP was 97.2%. There were 4 false negatives with RT-PCR Ct values of >30. How many samples with Ct values >30 were tested, or did all the other samples have low Ct values?  It would be necessary to calculate the sensitivity of RT-LAMP for different Ct value ranges. Otherwise, sensitivity of RT-LAMP would be overestimated.

Author Response

We thank the reviewer for his/her time to consider the manuscript and to provide us with the valuable comments. We have addressed the points raised by the reviewer below as follows:

In this manuscript Menting et al. evaluated an RT-LAMP assay for SARS-CoV-2. There are numerous papers describing the performance of RT-LAMP compared to RT-PCR, considering its advantages and limitations. The major shortcoming of this report is that it does not add relevant new information to the field. The authors state that there is a need for more accessible molecular tests for diagnostics in low- and middle-income countries with limited resources. Why then rely on extracted RNA rather than direct sample testing, which is possible with LAMP? What is the advantage of this protocol over PCR? The authors calculated that the sensitivity of RT-LAMP was 97.2%. There were 4 false negatives with RT-PCR Ct values of >30. How many samples with Ct values >30 were tested, or did all the other samples have low Ct values?  It would be necessary to calculate the sensitivity of RT-LAMP for different Ct value ranges. Otherwise, sensitivity of RT-LAMP would be overestimated.

Response: we think that this manuscript does add new information to the field as it reports on an independent diagnostic evaluation of a new test. To our opinion such is an evaluation is necessary for each test that becomes available on the market to ensure the quality of diagnostic testing. RNA extraction was done to allow testing with RT-PCR of the same sample. In total 9 samples had Ct>30 of whom 4 were negative with LAMP (this information is now added in the results section). Information on Ct values has been added in the text. The sensitivity presented for different Ct values does to our opinion not add much to the manuscript; we think it is sufficiently presented.

Reviewer 3 Report

The paper is very straightforward and the objectives of the study are clearly stated and described. The paper is clearly written and is of interest for researchers/clinicians in LMICs concerned with SARS-CoV-2 diagnostics. There are just very few minor changes to be made in the text:

1. Add "and" in "low-and-Middle Income Countries" in the first sentence of the Introduction.

2. Put "amplification" with a capital "A" in "Loop-mediated isothermal Amplification" in the first sentence of the second paragraph in the Introduction.

3. Line 50: The aim of the current study "was".

4. Line 73 clarify that the PCR assay was conducted using the same conditions as in the reference cited.

5. Wherever the gold standard SARS-CoV-2 PCR assay is mentioned I think it should be clarified that it is a quantitative PCR (qPCR) assay and not a conventional assay (so change RT-PCR or PCR with RT-qPCR or qPCR in the text where appropriate).

Author Response

We thank the reviewer for his/her time to consider the manuscript and to provide us with the valuable comments. We have addressed the points raised by the reviewer below as follows:

The paper is very straightforward and the objectives of the study are clearly stated and described. The paper is clearly written and is of interest for researchers/clinicians in LMICs concerned with SARS-CoV-2 diagnostics. There are just very few minor changes to be made in the text:

  1. Add "and" in "low-and-Middle Income Countries" in the first sentence of the Introduction.

Response: text has been modified according to the reviewer’s suggestion.

  1. Put "amplification" with a capital "A" in "Loop-mediated isothermal Amplification" in the first sentence of the second paragraph in the Introduction.

Response: text has been modified according to the reviewer’s suggestion

  1. Line 50: The aim of the current study "was".

Response: text has been modified according to the reviewer’s suggestion

  1. Line 73 clarify that the PCR assay was conducted using the same conditions as in the reference cited.

Response: we have modified the text as follows  “ which was performed complying to the established protocol [2].  “ and the reference to the protocol is mentioned (reference 2: Corman VM, Landt O, Kaiser M, Molenkamp R, Meijer A, Chu DK, Bleicker T, Brünink S, Schneider J, Schmidt ML, Mulders DG, Haagmans BL, van der Veer B, van den Brink S, Wijsman L, Goderski G, Romette JL, Ellis J, Zambon M, Peiris M, Goossens H, Reusken C, Koopmans MP, Drosten C. Detection of 2019 novel coronavirus (2019-nCoV) by real-time RT-PCR. Euro Surveill. 2020 Jan;25(3):2000045. doi: 10.2807/1560-7917.ES.2020.25.3.2000045

  1. Wherever the gold standard SARS-CoV-2 PCR assay is mentioned I think it should be clarified that it is a quantitative PCR (qPCR) assay and not a conventional assay (so change RT-PCR or PCR with RT-qPCR or qPCR in the text where appropriate).

Response: Indeed the gold standard assay is a real-time reverse transcription PCR (and not a conventional PCR). For correctness we have now wherever the gold standard assay is mentioned indicate that this is a real-time RT PCR.

Reviewer 4 Report

The authors compared a commercially available RT-LAMP kit with gold standard RT-PCR on SARS CoV2 detection using 55 positive and 55 negative SARS CoV2 samples. The RT-LAMP assay showed a good sensitivity and specificity, and good positive and negative predictive values.

I think the manuscript demonstrated a good performance of RT-LAMP. How ever I have some concern about the manuscript,

Comment

The authors did not describe the information of the gold standard RT-PCR assay that was used to test the samples.

The gold standard RT-PCR that was used in this study was a realtime reverse transcriptase or reverse transcriptase PCR assay?

Please provide information whether the gold standard RT-PCR assay was a commercially available assay or an in-house assay. Please provide the sensitivity and specificity of the RT-PCR assay used in this study.

Please describe about the differences in the running cost of RT-LAMP and RT-PCR and discuss the cost efficiency.

If the CT value of all the 55 positive samples are available, it should be describe in the manuscript.

Author Response

We thank the reviewer for his/her time to consider the manuscript and to provide us with the valuable comments. We have addressed the points raised by the reviewer below as follows:

The authors compared a commercially available RT-LAMP kit with gold standard RT-PCR on SARS CoV2 detection using 55 positive and 55 negative SARS CoV2 samples. The RT-LAMP assay showed a good sensitivity and specificity, and good positive and negative predictive values.

I think the manuscript demonstrated a good performance of RT-LAMP. How ever I have some concern about the manuscript,

Comment

The authors did not describe the information of the gold standard RT-PCR assay that was used to test the samples.

Response: we have now added that this is a real-time reverse transcription PCR and that it was performed according to reference 2 (Corman VM, Landt O, Kaiser M, Molenkamp R, Meijer A, Chu DK, Bleicker T, Brünink S, Schneider J, Schmidt ML, Mulders DG, Haagmans BL, van der Veer B, van den Brink S, Wijsman L, Goderski G, Romette JL, Ellis J, Zambon M, Peiris M, Goossens H, Reusken C, Koopmans MP, Drosten C. Detection of 2019 novel coronavirus (2019-nCoV) by real-time RT-PCR. Euro Surveill. 2020 Jan;25(3):2000045. doi: 10.2807/1560-7917.ES.2020.25.3.2000045.)

The gold standard RT-PCR that was used in this study was a realtime reverse transcriptase or reverse transcriptase PCR assay?

Response:  it is a real-time reverse transcription PCR and that it was performed according to reference 2 (see above).

Please provide information whether the gold standard RT-PCR assay was a commercially available assay or an in-house assay. Please provide the sensitivity and specificity of the RT-PCR assay used in this study.

Response: this information is now provided in lines 73-76.

Please describe about the differences in the running cost of RT-LAMP and RT-PCR and discuss the cost efficiency.

Response: we made a general  comment on costs already in the discussion (see line: 170). A formal cost effectiveness analysis was outside the scope of this manuscript.

If the CT value of all the 55 positive samples are available, it should be describe in the manuscript.

Response: we have now provided the mean Ct value of the positive samples with the minimum and maximum values (see lines: 130-131).

Round 2

Reviewer 2 Report

I have to repeat my criticism from the first review. There is no significant new information and the sample size is too small.

Reviewer 4 Report

The authors has respond to the reviewer's comment appropriately.